The function of small RNA in Pseudomonas aeruginosa

Liu Pei 1
Yue Changwu 1
Liu Lihua 2
Gao Can 1
Lyu Yuhong 1
Deng Shanshan 2
Tian Hongying feifei19890621@163.com 1
Jia Xu jiaxucmc@qq.com 2 3
1 Yan’an University, Key Laboratory of Microbial Drugs Innovation and Transformation , Yan’an , Shaanxi , China
2 Chengdu Medical College, Non-coding RNA and Drug Discovery Key Laboratory of Sichuan Province , Chengdu , Sichuan , China
3 School of Basic Medical Science, Chengdu Medical College , Chengdu , Sichuan , China
Jose Leny
Electronic publication date: 2022 Jul 21
Publication date: 2022
Volume: 10
Electronic Location ID: e13738
Received 2022 Apr 4; Accepted 2022 Jun 25
Copyright: ©2022 Liu et al.
Copyright year: 2022
Copyright holder: Liu et al.
License: This is an open access article distributed under the terms of the Creative Commons Attribution License, which permits unrestricted use, distribution, reproduction and adaptation in any medium and for any purpose provided that it is properly attributed. For attribution, the original author(s), title, publication source (PeerJ) and either DOI or URL of the article must be cited.
License URL: https://creativecommons.org/licenses/by/4.0/

Keywords: Pseudomonas aeruginosa, Small RNA, Post-transcriptional regulation, Drug targets, Antimicrobial resistance

Funding: National Natural Science Foundation of China 32170119 31870135 This work was supported by the National Natural Science Foundation of China (No. 32170119, 31870135) The funders had no role in study design, data collection and analysis, decision to publish, or preparation of the manuscript.

==============================
Pseudomonas aeruginosa, the main conditional pathogen causing nosocomial infection, is a gram-negative bacterium with the largest genome among the known bacteria. The main reasons why Pseudomonas aeruginosa is prone to drug-resistant strains in clinic are: the drug-resistant genes in its genome and the drug resistance easily induced by single antibiotic treatment. With the development of high-throughput sequencing technology and bioinformatics, the functions of various small RNAs (sRNA) in Pseudomonas aeruginosa are being revealed. Different sRNAs regulate gene expression by binding to protein or mRNA to play an important role in the complex regulatory network. In this article, first, the importance and biological functions of different sRNAs in Pseudomonas aeruginosa are explored, and then the evidence and possibilities that sRNAs served as drug therapeutic targets are discussed, which may introduce new directions to develop novel disease treatment strategies.

Introduction

In this review, we mainly focus on the biological functions and research progress of Pseudomonas aeruginosa small RNAs. We hope that clarifying the function of sRNAs will help to formulate new disease treatment strategies, and it may also lead to find new antibiotics, or new targets of existing antibiotics.

Survey methodology

A large number of documents (including clinical trials and reviews) on PubMed through the Internet were searched, which were then categorized and read meticulously. The key words are: Pseudomonas aeruginosa, small RNA. The first aspect of the inclusion criteria is that the article has complete structure and sufficient materials, and the other is that it contains retrieval keywords.

Background

Pseudomonas aeruginosa is a gram-negative conditional pathogen widely distributed in nature. People with low immunity (such as post-operative people (Belusic-Gobic et al., 2020; Nowikiewicz et al., 2020; Zhang et al., 2015) and HIV patients (Sonnleitner et al., 2020)) are susceptible, resulting in blood flow infection, respiratory infection, etc. In practice, there are not only single bacterial infections of Pseudomonas aeruginosa, but also co-infection with other bacteria. Different kinds of bacteria promote each other’s survival through nutritional cooperation to form chronic infection (Camus et al., 2020; Peng et al., 2020). When the infection is caused by the coexistence of Pseudomonas aeruginosa and Staphylococcus aureus, P. aeruginosa changes its own genotype and phenotype, which reduces its antibacterial effect on Staphylococcus aureus (Limoli et al., 2017). The whole-genome sequencing of P. aeruginosa further revealed its inherent resistance to antibiotics and strong environmental adaptability (Erdmann et al., 2018; Stover et al., 2000). Another study found a large number of gene mutations in the genome of P. aeruginosa in bacteria isolated from patients with cystic fibrosis (CF). This bacteria’s adaptive strategy can reduce the genome size and avoid the host immune response and the effect of antibiotics (Gabrielaite et al., 2020).

Small regulation RNA (sRNA) is one of the important means for bacteria to adapt to environmental changes and is involved in post-transcriptional regulation, such as adaptation to stress, virulence, and biofilm formation (Jorgensen, Pettersen & Kallipolitis, 2020). Most sRNAs are between 70–140 nt in length, usually primary transcripts, and sometimes may come from the 3′ terminal processing of longer mRNA precursors (Bossi et al., 2020). sRNA interacts with different target RNAs or proteins to affect their activity and function to regulate gene expression, which usually requires the participation of RNA chaperones such as Hfq and ProQ (Dutta & Srivastava, 2018). The maturation and degradation of sRNAs are related to the action of ribonuclease (Baek et al., 2019; Saramago et al., 2014). In another research, using high-throughput cDNA sequencing (RNA-seq), more than 500 new sRNAs were identified, significantly increasing the number of sRNAs found in P. aeruginosa (Gomez-Lozano et al., 2012). The present study only recognized the functions of some sRNAs, but little is known about the regulatory networks of these sRNAs and the functions of other uninvestigated sRNAs. This review will mainly shed light on the currently known sRNAs in P. aeruginosa with explanation of their biological functions and the recent research progress, as well as the prospect of selected sRNAs as direct or indirect targets for developing new drug therapies.

sRNA Classification in Bacteria

The sRNA can be divided into three classes according to their different functions. (1) The sRNAs that are base pairing to mRNAs. sRNAs regulate mRNAs post-transcriptionally, binding near ribosome binding sites (RBS) to inhibit its translation initiation or stimulate mRNAs decay. Alternatively, sRNAs may stimulate translation initiation or prevent mRNA degradation by base pairing to the 5′-UTR far upstream from the RBS, in which sRNAs can be divided into cis encoded sRNAs and trans encoded sRNAs. Cis-encoded sRNAs are transcribed from the DNA strand and are complementary to target mRNAs. Trans encoded sRNA is transcribed from a completely different genomic location from the gene of its target mRNA. As sRNA is less complementary to the target mRNA, in most cases, they require the assistance of chaperones to facilitate sRNA-mRNA stability interactions. Limited complementarity allows trans encoded sRNAs to base-pair with multiple targets (Dutta & Srivastava, 2018; Jorgensen, Pettersen & Kallipolitis, 2020). (2) Protein-targeted sRNAs. sRNAs regulate the expression of many genes indirectly by sequestering proteins, inhibiting these proteins regulatory functions (Dutta & Srivastava, 2018). (3) sRNAs associated with CRISPRs (clustered regularly interspaced short palindromic repeats). CRISPR-derive RNAs (crRNAs) are a short stretch of RNAs against foreign nucleic acids, and their main role is to guide the nuclease Cas to bind exogenous nucleic acids, thereby exerting the function of CRISPR-Cas system to clear exogenous nucleic acids (Behler & Hess, 2020). CRISPR-Cas systems exist in many prokaryotes, for example, in Listeria, the non-coding RNA RliB is an atypical member of the CRISPR family, which can regulate phage interactions with host strains (Sesto et al., 2014; Tian et al., 2021).

Biological Functions of sRNAs in Pseudomonas aeruginosa

Carbon, nitrogen, and iron metabolism

P. aeruginosa is an opportunistic pathogen with strong environmental adaptation (Jurado-Martín, Sainz-Mejías & McClean, 2021), which developed a complex metabolic network during a long period of evolution (Dolan et al., 2020; Rossi et al., 2021). Two specialized two-component regulatory systems (TCS), CbrA/CbrB and NtrB/NtrC of P. aeruginosa, are important parts of the sensing and response to nutrients in the environment by discerning the same or interrelated signal types (Nishijyo, Haas & Itoh, 2001). The CbrA/B system is involved in carbon source utilization and carbon catabolic repression (CCR) through activation of the sRNA CrcZ in P. aeruginosa (Valentini et al., 2014). The NtrB/C two-component system is an important regulator of nitrogen assimilation and cluster motility in P. aeruginosa. Under nitrogen deficiency, which NtrB/C acts synergistically with RpoN to induce sRNA NrsZ production (Wenner et al., 2014). A study showed that prrF encoded sRNA is required to maintain iron homeostasis during infection by P. aeruginosa (Reinhart et al., 2017), while iron regulatory pathways are altered in P. aeruginosa under static growth conditions (Brewer et al., 2020). Moreover, the sRNA PA2952.1 and PrrH also regulate iron metabolism (Coleman et al., 2021).

Biofilm formation

The biofilm of P. aeruginosa consists of bacteria, extracellular DNA (eDNA) (Seviour et al., 2021), proteins, rhamnolipids (a biosurfactant with antibacterial activity involved in surface motility and biofilm formation) (Abdel-Mawgoud, Lépine & Déziel, 2010; Ali et al., 2021) and extracellular polysaccharides (PSL, PEL, alginate) (Moradali, Ghods & Rehm, 2017). In the growth state of biofilm, P. aeruginosa can resist the action of multiple adverse environments, significantly improving the ability of bacteria to resist antibiotics (Thi, Wibowo & Rehm, 2020). The sRNA ErsA promotes biofilm development through AmrZ (alginate and motility regulator Z) post-transcriptional regulation (Falcone et al., 2018). As a global transcription regulator, AmrZ participates in the regulation of biofilm and virulence of P. aeruginosa (Xu et al., 2016). In P. aeruginosa biofilms the sRNA SrbA is detected to be highly upregulated (Taylor et al., 2017). Similarly, many sRNAs are involved in the regulation of biofilm formation, including RsmZ, RsmY, RsmW, RsmV, PhrS, ReaL, PrrH, NrsZ, PhrD, and Pa2952.1 (Fig. 1). Their specific regulation mechanisms are shown in section 3.

Figure 1 Regulation mechanisms of various small RNAs in Pseudomonas aeruginosa on biofilm.

The irregular light green figure in the middle of this picture shows the biofilm of Pseudomonas aeruginosa. ① BswR requires GacA to upregulate rsmZ. ② BswR may act by counteracting the repressor MvaT in upregulation of rsmZ. ③ AmrZ binds to the algD promoter (Xu et al., 2016 and Xu et al., 2016). ④ AmrZ modulates Pseudomonas aeruginosa biofilm by directly repressing transcription of the psl operon (Jones et al., 2013). ⑤ Crcz participates in biofilm formation by competing Hfq with other sRNAs.

Quorum sensing

Quorum sensing (QS) is an intercellular signal communication system based on small signal molecules. P. aeruginosa controls virulence and biofilm formation through quorum sensing system (O’Loughlin et al., 2013) to regulate the transformation between bacterial planktonic state and biofilm state. QS is regulated hierarchically, which consists of interconnected las, rhl, pqs, and iqs systems (Malgaonkar & Nair, 2019). LasR and RhlR control the key virulence factors (O’Loughlin et al., 2013).The las system is at the top of QS hierarchical network. The complex of LasR (QS related regulator) combined with signal molecule 3-oxo-C12HSL can regulate the other three systems which are RhlR, PqsR and IqsR (QS related regulator). These three systems regulate other pathways when combined with corresponding signal molecules (C4HSL, PQS and IQS) (Lee & Zhang, 2015; Passos da Silva et al. 2017; Soukarieh et al., 2018). It is worth noting that sRNAs also play an important role in QS regulation system (Fig. 2). The sRNA ReaL function is to connect the las and pqs systems (Carloni et al., 2017). The rhl system is positively regulated by sRNA PhrD and sRNA RhlS, while negatively regulated by sRNA p27 (Chen et al., 2019; Malgaonkar & Nair, 2019; Thomason et al., 2019), and also RhlI (QS related regulator) negatively regulates the level of PrrH (Lu et al., 2019). sRNA PrrF1/2 regulates PQS synthesis by inhibiting antR (Djapgne et al., 2018). RsmZ/Y participates in the regulation of QS by antagonizing RsmA protein. RsmA is a regulatory protein that negatively regulates the production of extracellular product pyocyanin (a blue–green pigment that can interfere with host cell redox reactions (Lau et al., 2004)) as well as quorum sensing signaling molecules C4HSL and 3-oxo-C12HSL, and also RsmA positively regulates swarming (a complex mode of motion that causes bacteria to form tendrils on semisolid surfaces (Caiazza, Shanks & O’Toole, 2005)) and rhamnolipid synthesis (Heurlier et al., 2004; Pessi et al., 2001).The sRNA PhrS acts as an activator of PqsR synthesis, which is stimulated the oxygen response regulator Anr (a global anaerobic response regulator) (Sonnleitner et al., 2011).

Figure 2 Small RNAs involved in quorum sensing regulation.

① CrcZ participates in the regulation of antR by competing with PrrF1 / 2 for Hfq.

Drug resistance

P. aeruginosa can become drug-resistant strains by genetic mutations and horizontal transmission of resistance genes within themselves (Botelho, Grosso & Peixe, 2019). For example, outer membrane porin oprD mutations and overexpression of the native β-lactamase ampC are responsible for carbapenem resistance, and overexpression of the efflux pumps mexX and mexA is associated with resistance to aminoglycosides and carbapenems, respectively (Aghazadeh et al., 2014; Feng et al., 2021). Current study found that at least six sRNAs are involved in the regulation of drug resistance in P. aeruginosa, and there are differences in the regulatory mechanisms of different sRNAs. These mechanisms are as follows. The sRNA AS1974 is a major regulator to control the expression of multiple resistance pathways, including membrane transporters and biofilm-associated antibiotic resistance genes. The sRNA AS1974 can transform drug-resistant strains into antibiotic sensitive ones (Law et al., 2019). TpiA (triose phosphate isomerase) influences aminoglycoside antibiotic resistance via sRNA CrcZ (Xia et al., 2020a). In another study, when overexpressing sRNA PA0805 1 and sRNA 2952.1, the expression of mexGHI-opmD, a drug efflux system, was up-regulated and as a result, the bacterial resistance to aminoglycoside antibiotics increased (Coleman et al., 2021; Coleman et al., 2020). ErsA and sRNA Sr0161 increase bacterial resistance to carbapenems by inhibiting the translation of oprD (Zhang et al., 2017). Bacterial resistance to polymyxins increases following base complementary pairing of sRNA Sr006 with pagL (an enzyme responsible for deacylation of lipid A) mRNA (Zhang et al., 2017).

Virulence factors

P. aeruginosa has different virulence factors in acute infection and chronic infection. There are several virulence factors for acute infection: flagella, type IV pili, lipopolysaccharide, exotoxin A, ectoenzyme S, type III section system (T3SS), and so on (Ben Haj Khalifa et al., 2011). The T3SS is a bacterial secretory channel capable of injecting different effectors into host cells to influence host immune mechanisms and provide a favorable environment for bacterial survival (Horna & Ruiz, 2021; Lombardi et al., 2019). Expression of T3SS is associated with several proteins, including ExsA and Vfr, which are two DNA binding proteins (Urbanowski, Lykken & Yahr, 2005). Vfr promotes T3SS expression by activating the PexsA promoter (Marsden et al., 2016). The sRNA 179 is an Hfq dependent repressor of T3SS gene expression while it also inhibits ExsA and Vfr synthesis (Janssen et al., 2020). Experimental studies have found that overexpression of the sRNA PA2952.1 leads to impaired P. aeruginosa motility (downregulation of pilus and flagella gene expression), decreased cytotoxicity detected in PrrH deleted mutants, and increased P. aeruginosa siderophore production (Coleman et al., 2021). ReaL bases pairing the sequence of SD sequence of rpoS mRNA, making it silent without translation process. RpoS (σS) is involved in quorum sensing and the regulation of several virulence genes (Thi Bach Nguyen et al., 2018). Whereas loss of ReaL impaired the virulence phenotype of P. aeruginosa, overexpression of ReaL resulted in a hypervirulent phenotype (Carloni et al., 2017). With the condition of anaerobic growth and 37 °C, production of sRNA PesA (present only in P. aeruginosa PA14 strain) was induced, which strengthens bacterial virulence while promoting pyocyanin S3 synthesis (Ferrara et al., 2017).

As there are a lot of investigations focused on sRNAs, we have found that the biological functions of most sRNAs are not single, moreover, some sRNAs appear to function as global regulators in post-transcriptional regulatory networks. For instance, by overexpressing sRNA PA0805.1 in P. aeruginosa wild-type (WT) PAO1, many phenotypes (including motility, cytotoxicity, and drug resistance) were found to be altered, making it probable that sRNA PA0805.1 is a global regulator (Coleman et al., 2020). Although the depth and breadth of P. aeruginosa sRNAs research are currently increasing, knowledge of the specific regulatory mechanisms of various sRNAs is lacking. Understanding the current state of sRNA research is a prerequisite for further elucidation of the complex post-transcriptional regulatory mechanisms. Next, the characteristics and functions of various sRNAs will be described in detail (Table 1).

Table 1 summary of the biological functions of eleven sRNAs.

sRNA	Transcript length	Gene location	Whether Hfq dependent	Target	Function	
RsmZ	116nt	PAO1_4,057,543-4,057,658	Not describe	RsmA/F	Associated with biofilm formation, motility, and expression of T3SS.	
RsmY	124nt	PAO1_586,867-586,990	Yes	RsmA/F	Associated with motility, and the expression of T3SS.	
RsmW	224nt	PAO1_5,117,971-5,118,195	Not describe	RsmA/F	Associated with biofilm formation.	
RsmV	192nt	PAO1_1011621-1011812	Not describe	RsmA/F	Sequestration of RsmA/F from target mRNAs; activates translation of the T6SS component tssA1; represses the expression of the T3SS gene.	
PrrF1/2	151/148nt	PAO1_5,283,960-5,284,110/PAO1_5,284,172-5,284,319	Yes	antR mRNA	Expression of the sRNA PrrF1/2 is regulated by Fur, which is associated with iron homeostasis, heme balance, biofilm formation, expression of virulence genes, twitching motility, and synthesis of PQS.	
PrrH	325nt	PAO1_5283995-5284319	Yes	nirL	Involved in the regulation of heme, quorum sensing and bacterial virulence.	
PhrS	213nt	PAO1_3,705,309-3,705,521	Yes	pqsR	Regulated by ANR, PhrS stimulates the translation of pqsR and promotes the synthesis of PQS and PYO, which are involved in biofilm formation.	
NrsZ	226nt	PAO1_5775397-5775623	Not describe	rhlA	Regulated by the cooperation between NtrB/C and RpoN; involved in the regulation of swarming motility.	
RgsA	197nt	PAO1_3,318,663-3,318,859	Yes	rpoS mRNA; fis mRNA; acpP mRNA	Regulated by GacA and RpoS; involved in oxidative stress response, affecting bacterial virulence and motility.	
ReaL	100nt	PAO1_3958000-3958200/ PA14_1599900-1600100	Yes	pqsC; rpoS mRNA	Negatively regulated by lasR; promotes the synthesis of PQS; correlates with bacterial virulence expression.	
ErsA	130nt	PAO1_6183500-6183700/ PA14_6456400-6456600	Yes	algC mRNA; oprD mRNA; amrZ mRNA	Regulated in response to envelope stress; affects biofilm formation; involved in regulating the expression of bacterial AlgC enzyme, drug resistance and motility.	

Properties and Functions of Different sRNAs

sRNAs acting on RsmA/F proteins

In P. aeruginosa, Rsm (repressor of stationary-phase metabolites, Rsm) protein family has been proved to play an important role in post-transcriptional regulation. Rsm protein family are involved significantly role in the bacterial response to environmental changes by binding to target mRNA to effectively inhibit or promote protein translation (Potts et al., 2017). There are four different sRNAs that can bind to RsmA / F and then isolate RSMA / F from target mRNAs (Janssen et al., 2018a).

sRNA RsmZ and sRNA RsmY

RsmZ sRNA is encoded by a prrB related gene which exists in the form of 127 nucleotide RNA in cells, and has an affinity for RsmA protein (Heeb, Blumer & Haas, 2002). In vitro, it is found that the integrated host factor (IHF) protein had a high affinity with the rsmZ promoter region, suggesting that DNA bending was involved in regulating rsmZ expression. The expression of rsmZ requires GacA protein which is a global activator. GacA is closely related to the Pseudomonas quorum sensing system and biofilm formation (Reimmann et al., 1997). The expression of rsmZ also needs promoter with highly conserved UAS which is a conserved palindrome upstream activation sequence TGTAAG…CTTACA (Humair, Wackwitz & Haas, 2010; Kay et al., 2006).

RsmY gene is located between dnr gene of P. aeruginosa PAO1 and open reading frame of PA0528. The transcription of rsmY and rsmZ is positively regulated by RsmA while negatively regulated by RsmY and RsmZ. However, when rsmY and rsmZ genes coexist, the transcription of RsmY or RsmZ is inhibited (Kay et al., 2006). The rsmY transcription is activated by the GacS/GacA two-component system. The secondary structure of RsmY is similar to RsmZ (Fig. 3). The transcript of rsmY is about 120 nt., which has the highest content in the stable phase and can interact with the translation regulator RsmA (Valverde et al., 2003).

Figure 3 RsmY has a secondary structure similar to RsmZ.

(A) Predicted P. aeruginosa RsmZ secondary structure determined by Mfold modeling. (B) SHAPE-MaP structure for P. aeruginosa RsmZ. (C) Predicted Mfold structure for P. aeruginosa RsmY. (D) SHAPE-MaP structure for P. aeruginosa RsmY (Janssen et al., 2018a; Janssen et al., 2018b). Copyright ©  2018 American Society for Microbiology.

RsmA has two preferential binding sites on RsmY and RsmZ, while RsmF has one preferential binding site on RsmY and two preferential binding sites on RsmZ. RsmF has higher binding conditions both in vivo and in vitro (Janssen et al., 2018b).

GacS/GacA two-component system positively controls the expression of the quorum-sensing system and extracellular products through two small regulatory RNAs RsmY and RsmZ, which affect biofilm formation (Kay et al., 2006). Environmental changes can upregulate the expression of RsmY and RsmZ to increase bacterial population density and population defense (Zhao et al., 2014).

RsmY and RsmZ interact with other sRNAs during regulation, for example, expression of sRNA 179 stimulate transcription of RsmY, and both RsmY and RsmZ are required for sRNA 179 to regulate T3SS gene expression: sRNA 179 indirectly affects translation of ExsA by modulating RsmY levels, thereby affecting RsmA utilization (Janssen et al., 2020). The sRNA RsmY and RsmZ are in a complex regulatory network. In another study, SuhB (a regulator of multiple virulence genes (Li et al., 2013)) negatively regulates motility and biofilm formation through GacA-RsmY/Z-RsmA cascade. Mutations in GacA or two sRNAs RsmY and RsmZ, or overproduction of RsmA protein, basically improved the motility defect of suhB mutant (Li et al., 2017). Micro-aerobic environment significantly inhibited the expression of sRNA RsmY and RsmZ, which was mediated by NarL, an anaerobic response regulator regulated by Anr (O’Callaghan et al., 2011). RsmZ is also affected by transcription regulators. For instance, BswR (bacterial swarming regulator) can counteract the repressive activity of MvaT (H-NS-like DNA-binding protein), as well as control the transcription of RsmZ. In addition, BswR can regulate the biogenesis of bacterial flagella, and play an important role in regulating the movement and the formation of biofilm in P. aeruginosa (Wang et al., 2014).

sRNA RsmW

RsmW is a RsmY/RsmZ type sRNA derived from PA4570 3′- UTR. The RNA-seq show higher levels of RsmW and greater stability of RsmW compared to PA4570, but it is not certain whether the RsmW sRNA is an independent transcriptional event. The secondary structure of RsmW is highly similar to RsmZ and RsmY, and RsmW contains seven GGA motifs (a special sequence consisting of three consecutive ribonucleotides on RNA), three of which are exposed in a single-stranded outer stem-loop, suggesting that it is involved in the regulation of RsmA and RsmA can regulate PA4570 and RsmW transcript levels. The affinity of RsmW for RsmA (Kd = 11.5 ± 1.5 nm) is higher than that of RsmY for RsmA (Kd = 55 ± 7 nm) (Sonnleitner et al., 2006). RsmW levels increased with increasing temperature, and also its expression was up-regulated during biofilm growth. Compared with wild type, RsmW expression was enhanced in the logarithmic growth phase and late stationary phase in gacA or the rhlR transposon mutant. In mutants which both RsmY and RsmZ are deleted, RsmW can compensate for the loss of RsmY and RsmZ and promote biofilm formation (Miller et al., 2016).

sRNA RsmV

RsmV, a transcript of 192 nt, is highly conserved in the genome of P. aeruginosa with four predicted RsmA/RsmF consensus binding sites-four CANGGAYG (GGA2, GGA3, GGA5, GGA6) sequences in a stem-loop structure. Each CANGGAYG sequence contributes to RsmV activity. RsmV can sequester RsmA and RsmF from target mRNAs in vivo to activate translation of tssA1, which is a component of the type VI secretion system (T6SS, can inject effector proteins into eukaryotic cells (Allsopp et al., 2017)). Followed by tssA 1 activation, T3SS gene expression was repressed. All of sRNAs RsmV, RsmW, RsmY, and RsmZ have the ability to sequester RsmA and RsmF. Still, sRNAs may play different roles in the sequestration of RsmA/RsmF depending on their expression timing (Janssen et al., 2018a), which may be related to the mechanism that fine-tunes the Rsm system in response to changes in the external environment.

sRNA PrrF1 and sRNA PrrF2

PrrF1 and PrrF2 sRNAs, functional homologs of RyhB sRNAs in E. coli, are part of the regulatory network of iron metabolism in P. aeruginosa, which affect the expression of at least 50 genes encoding iron-containing proteins (Reinhart et al., 2015). The tandemly encoded sRNA PrrF1 and sRNA PrrF2 are more than 95% similar to each other, while a functional Fur box precedes each sRNA. Fur is a transcriptional repressor to regulate iron uptake by regulating the expression of sRNA PrrF1 and sRNA PrrF2, which was induced under conditions of iron deficiency. PrrF1 and PrrF2 have overlapping effects on regulating genes, including iron storage, antioxidant stress, and intermediate metabolism (Wilderman et al., 2004).

By constructing prrf deficient mutant strains, the researchers found that iron homeostasis, heme balance, biofilm formation, and virulence gene expression were affected, among which the most significant change is the decrease of bacterial virulence (Reinhart et al., 2015). During acute lung infection, sRNA PrrF is necessary to maintain iron homeostasis and virulence during the growth of P. aeruginosa (Reinhart et al., 2017). PhuS is mainly a heme-binding protein. In addition to playing a role in extracellular heme metabolism, PhuS can also act as a transcriptional regulator to regulate the levels of PrrF and PrrH in response to heme changes. This dual function of PhuS helps to integrate the utilization of extracellular heme into the PrrF / PrrH sRNAs regulatory network, which is very important for the adaptability and virulence of P. aeruginosa (Wilson, Mourino & Wilks, 2021).

PrrF1/2 sRNAs are also involved in the regulation of quorum sensing. PrrF represses the gene encoding the anthranilate degrading enzyme (i.e., antABC), a precursor of the Pseudomonas quinolone signal (PQS). PrrF RNA inhibits the degradation of anthranilic acid in an iron-deficient environment, allowing biosynthesis of PQS (Oglesby et al., 2008). PrrF1/2 sRNAs promote the production of AQS (2-akyl-4 (1H) - quinolone metabolites) by repressing the translation of antR, which encodes transcriptional activators of anthranilic acid degradation genes. AQS mediates a range of biological activities, including quorum sensing and inter bacterial interactions. PrrF sRNA interacts with the antR mRNA 5′-UTR (Djapgne et al., 2018) with Hfq stabilizing the structure of PrrF sRNAs and stimulates base pairing between the sRNA PrrF and the antR mRNA (Sonnleitner, Prindl & Blasi, 2017).

In a novel study, PrrF sRNAs were shown to be involved in regulating the twitching motility, during iron limited-conditions, which is a motion pattern using type IV pili moving on moist surfaces (Mattick, 2002; Nelson et al., 2019). The iron regulatory pathway of P. aeruginosa is altered in a static growth state. The HSI-II T6SS site is a novel PrrF responsive system, in which PrrF regulates T6SS gene expression under static conditions by promoting AQ production (Brewer et al., 2020). These studies confirm that PrrF1 and PrrF2 are essential in the physiology and pathogenesis of P. aeruginosa.

sRNA PrrH

The third full-length 325 nt transcripts, PrrH, encoded by the prrF locus, whose transcription starts at the 5 ’end of prrF1 and proceeds through the prrF1 terminator and the prrF1-prrF2 intergenic sequence (95 nt) while terminates at the 3′ end of the prrF2 gene. Expression of this transcript is repressed by heme and iron, with the most significant change in the stationary phase. The outer membrane heme receptors of PhuR and HasR play important roles in PrrH involved heme regulation. The nirL is a gene related to heme biosynthesis. The activation of nirL by iron and heme depends on prrF site, however, the regulation of nirL by heme is not due to the interaction between nirL mRNA and PrrF sRNAs, but PrrH’s regulating gene expression through its unique sequence from prrF1-prrF2 intervening region (Oglesby-Sherrouse & Vasil, 2010). PrrH was also shown to play a regulatory role in the quorum-sensing system. RhlI in the rhl system represses PrrH expression at the transcriptional level. PrrH directly inhibits LasI and PhzC / D, which is a part of a novel RhlI/R-PrrH-LasI/PhzC/PhzD signaling cascade that may be relevant to P. aeruginosa pathogenicity (Lu et al., 2019). PrrH affects pyocyanin and elastase production, which is the main component of the exocrine protein of P. aeruginosa and an important virulence factor for the pathogen to infect the host (Li et al., 2019). PrrH is also involved in rhamnolipid production, biofilm formation, swarming and motility in swimming, which is a motion pattern that utilizes flagella to swim in liquid (Yeung, Parayno & Hancock, 2012). All these functions indicate the importance of PrrH in bacterial virulence formation (Coleman et al., 2021; Lu et al., 2019).

sRNA PhrS

The phrS gene has an open reading frame (ORF) capable of encoding a 37 aa polypeptide, but whether the polypeptide has a recognizable physiological function remains to be elucidated (Sonnleitner et al., 2011). The sRNA PhrS, when overexpressed, was shown to be involved in nuclear transcriptional regulation. Thus PhrS appears to be a bifunctional sRNA that can act both as a nuclear transcriptional regulator and an mRNA (Sonnleitner et al., 2008). Synthesis of PhrS is highly up-regulated by the oxygen response regulator Anr, which is activated under hypoxia. PhrS is the first sRNA to provide a regulatory link between oxygen availability and quorum sensing, which may affect P. aeruginosa biofilm growth under hypoxia. The sRNA PhrS is involved in the regulation of quorum sensing. It is an activator of PqsR synthesis, while PqsR is one of the key regulators of quorum sensing in P. aeruginosa. A highly conserved region of 12 nucleotides located at the downstream of the internal open reading frame of phrS gene (169 to 182 nucleotides within the downstream of PhrS transcription initiation) is called the creg element of PhrS, which is necessary for uof (upstream open reading frame)—pqsR regulation. In this mechanism, PhrS promotes PQS and pyocyanin synthesis by stimulating pqsR translation (Sonnleitner et al., 2011). Moreover, PhrS is also an essential part of P. aeruginosa biofilm (Fengqin et al., 2017).

CRISPR-Cas is a prokaryotic adaptive immune system that protects phages and other parasites (Hoyland-Kroghsbo et al., 2017). The anti-termination effect mediated by PhrS promotes the transcription of CRISPR site to produce crRNA and makes CRISPR-Cas form acquired immunity to phage invasion. The regulation of the CRISPR system also requires the participation of PhrS creg motif (Lin et al., 2019).

sRNA NrsZ

NrsZ is encoded in the ntrC-PA5126 spacer region of PAO1, which is processed into two short transcripts of approximately 40 nt and 140 nt in response to nitrogen limitation. Because the expression of this sRNA is dependent on nitrogen source, it was named NrsZ (nitrogen regulated sRNA), which is produced as transcripts with at least 226 nt. NrsZ is induced under nitrogen limiting conditions by the NtrB/C two-component system in cooperation with RpoN. The transcriptional activity of the RpoN promoter was enhanced in a limited nitrogen source environment. NrsZ can regulate the swarming motility of P. aeruginosa. The first 60 nt of NrsZ containing SLI is a functional unit that regulates the swarming motility. NrsZ with conserved motif ACAGGCAG activates the expression of rhlA at the post-transcriptional level, which is an essential gene for rhamnolipid synthesis (Wenner et al., 2014).

sRNA RgsA

RgsA is a 120 nt sRNA controlled by GacA (Gonzalez et al., 2008). By constructing the rgsA deficient mutant of P. aeruginosa, it was found that the peroxide resistance of the bacteria diminished in both the planktonic and biofilm states, and the growth rate of P. aeruginosa was reduced, underscoring the important role of rgsA in the defense of P. aeruginosa against oxidative stress (Hou et al., 2021). Expression of RgsA requires the participation of RpoS (Gonzalez et al., 2008). RpoS activates the transcription of RgsA at each growth stage of bacteria. RgsA reduces the rpoS mRNA and RpoS protein levels at the post-transcriptional level for bacteria in the exponential growth stage, and this inhibition depends on Hfq (Lu et al., 2018). The mRNA encoding the global transcription regulators of Fis and acyl carrier protein AcpP are two direct regulatory targets of RgsA in P. aeruginosa. RgsA downregulates Fis and AcpP synthesis by base pairing with mRNA, a regulatory process requiring the participation of the highly conserved 71–77 region of RgsA and this regulation also needs the interaction site (141 to 175) at the downstream of the region. RNA chaperone Hfq is also required for this regulation. RgsA also affects motility and pyocyanin synthesis, suggesting an important role for RgsA in relevant processes involved in regulating virulence (Lu et al., 2016). Linking Fis to RpoS through RgsA has helped to reveal the complex interplay between sRNAs and transcriptional regulators. A study found that RgsA was down-regulated nearly four-fold in biofilms of mixed-species (S. aureus and P. aeruginosa) (Miller et al., 2017).

sRNA ReaL

ReaL is a transcript about 100 nt, and its level is affected by the temperature and available oxygen in the host. In the quorum sensing system, the sRNA ReaL is negatively regulated by the las regulator lasR (Carloni et al., 2017). Though, ReaL positively regulates the pqsC gene post-transcriptionally, thereby promoting the synthesis of PQS, and stimulating the connection between the las and pqs systems. ReaL also has a non-negligible function in P. aeruginosa pathogenic mechanisms: loss of ReaL leads to attenuated bacterial virulence, whereas ReaL overexpression results in a hypervirulent phenotype. ReaL affects pyocyanin synthesis, biofilm formation, and swarming motility, while these processes are all affected by PQS (Carloni et al., 2017).

YbeY is a highly conserved bacterial ribonuclease, and ReaL is the target of YbeY, which reduces sRNA ReaL levels. Increased levels of sRNA ReaL were found by constructing a YbeY deletion mutant (Xia et al., 2020b). In this study, overexpressed ReaL base pairs (Hfq dependent) with the SD sequence of rpoS mRNA to directly inhibit the translation of rpoS (Thi Bach Nguyen et al., 2018), thereby reducing the expression of oxidative stress-responsive genes (Xia et al., 2020b).

sRNA ErsA

ErsA consists of approximately 130 nt, which is upregulated by the changes of temperature (transition from ambient to host body temperature), and the changes in oxygen status (aerobic to anaerobic). ErsA is also transcriptionally regulated by the envelope stress response, which is controlled by σ22 activity, while σ22 activity affects P. aeruginosa pathogenicity (Ferrara et al., 2015). ErsA acts as a trans encoded sRNA that is currently known to bind to three mRNAs (Falcone et al., 2018; Ferrara et al., 2015; Zhang et al., 2017). One is through post-transcriptional negative regulation (Hfq dependent) of the algC gene encoding the virulence-associated enzyme AlgC, affecting exopolysaccharide production and biofilm formation (Ferrara et al., 2015). Like ErsA, activation of algC expression is dependent on σ22 (Xu et al., 2021), and thus ErsA and σ22 finely co-regulate AlgC enzyme expression in an incoherent feed-forward loop (Ferrara et al., 2015). Second, the base complementary pairing of the sRNA ErsA to the 5′-UTR of OprD mRNA leads to increased meropenem resistance in P. aeruginosa, in which OprD is responsible for carbapenem uptake (Zhang et al., 2017). Third, it binds to and positively regulates amrZ mRNA at the post-transcriptional level, to promote biofilm development, and to regulate bacterial swarming motility and twitching motility (Falcone et al., 2018). ErsA mediated regulation has been implicated in the pathogenicity of P. aeruginosa during the progression of acute infections. The regulation mechanism contributes to the stimulation of the host’s infected epithelial cells to initiate inflammatory responses. During CF chronic infection, adaptive mutations occur, which lead to downregulation of ErsA, enabling chronic colonization of the human lung by P. aeruginosa, possibly due to the action of selective pressure. As an important regulatory element in the interaction between host and pathogen, ErsA may contribute to the pathological adaptability of P. aeruginosa in the process of CF chronic infection in some cases (Ferrara et al., 2020). ErsA was upregulated approximately six-fold in biofilms of mixed species (S. aureus and P. aeruginosa) (Miller et al., 2017).

Table 2 Brief description of the biological functions of the other twelve different sRNAs.

sRNA	Transcript length	Gene location	Whether Hfq dependent	Target	function	
AS1974	127nt	PA185388(R3)_471298-471425	Yes	Not describe	Master regulator regulating multiple drug resistance pathways, including membrane transporters and biofilm associated drug resistance genes, the expression of which is regulated by gene 5′UTR methylation sites; it was able to transform multi drug resistant clinical strains into drug highly susceptible strains when overexpressed (Law et al., 2019).	
CrcZ	407nt	PAO1_5,308,587-5,308,993	Yes	Crc,Hfq	RpoN and CbrA/CbrB are required for crcZ expression. The CbrA-CbrB-CrcZ-Crc system enables bacteria to adapt to different carbon sources (Sonnleitner, Abdou & Haas, 2009). CrcZ binding to Hfq can sequester Hfq and affect multiple Hfq involved physiological activities: ① abolishes Hfq mediated translational repression of amiE mRNA (Sonnleitner & Blasi, 2014); ② indirectly affects biofilm formation by competing for Hfq (Pusic et al., 2016); ③ interferes with PrrF1-2/Hfq mediated regulation of the antR (Sonnleitner, Prindl & Blasi, 2017); ④ correlation with bacterial susceptibility to antibiotics (Pusic et al., 2018; Xia et al., 2020a; Xia et al., 2020b).	
P27	192nt	PAO1_4781786-4781978	Yes	rhlI mRNA	Fine tuning the activity of the rhl QS system (Chen et al., 2019).	
PA0805.1	276nt	PAO1_883,307–883,582	Not describe	Not describe	Associated with P. aeruginosa motility, adhesion, cytotoxicity and tobramycin resistance (Coleman et al., 2020; Gill et al., 2018).	
PA2952.1	117nt	PA14_3,312,577–3,312,693	Not describe	Not describe	PA2952. 1 affects P. aeruginosa virulence, motility, and antibiotic resistance, with links to several proteins and genes (Coleman et al., 2021; Gill et al., 2018).	
PaiI	126nt	PA14_13970-13990	Yes	Not describe	Induced in an anaerobic environment in the presence of nitrate, and transcription of PaiI is dependent on the two-component system NarX/L; PaiI has an important role in adaptive anaerobic denitrification (Tata et al., 2017).	
PhrD	73nt	PAO1_785,498-785,570	Yes	RhlR mRNA	Overexpression of PhrD increases the level of RhlR transcript, rhamnolipid and pyocyanin production; PhrD has a sequence specific promoting effect on RhlR transcripts without the involvement of any Pseudomonas specific proteins (Malgaonkar & Nair, 2019).	
RhlS	70nt	PAO1_3889700-3899900	Yes	fpvA mRNA	Complementary pairing with fpvA mRNA base to regulate its translation; when entering the stable phase, RhlS accumulates and produces normal levels of C4-HSL by stimulating RhlI mRNA translation (Thomason et al., 2019).	
Sr006	123nt	PAO1_182,570-182,693	Yes	pagL mRNA	Positively regulates the expression of PagL, reduces its pro-inflammatory properties and leads to polymyxin resistance (Zhang et al., 2017).	
Sr0161	247nt	PAO1_184,211–184,458	Yes	oprD mRNA	Base pairing with 5 ’UTR of OprD results in increased bacterial resistance to meropenem. Inhibits T3SS after interacting with exsA mRNA (Zhang et al., 2017).	
SrbA	239nt	PA14_2,977,373–2,977,611	Not describe	With a large number of different mRNA targets	SrbA plays an important role in biofilm formation and pathogenicity of P. aeruginosa (Gill et al., 2018; Taylor et al., 2017).	
sRNA52320	Not describe	Not describe	Not describe	Host mRNAs	sRNA52320 is rich in OMV (outer membrane vesicle), which can inhibit the secretion of IL-8 and KC cytokines induced by LPS and OMV, and reduce the infiltration of neutrophils in mouse lung. It participates in pathogen-host interaction and reduces host immune response (Koeppen et al., 2016).	

other sRNAs (Table 2)

The Possibility of sRNAs as Drug Targets

Small RNAs are inseparable from bacterial resistance or sensitivity to antibiotics by participating in the regulation of bacterial metabolism. sRNAs can be seen as a target of direct or indirect drug action, modulating bacterial susceptibility to antibiotics. Some sRNAs have been found to be closely related to the effectiveness of antibiotics. TpiA is a key enzyme affecting P. aeruginosa virulence and antibiotic resistance. In one of the studies of Yushan Xia et al. in 2020, it was found that TpiA is affecting P. aeruginosa virulence and aminoglycoside antibiotic resistance through sRNA CrcZ (Xia et al., 2020a). Using tobramycin to treat infections caused by Pseudomonas aeruginosa are prone to adaptive phenomena, and formation of biofilms. Increased expression of PrrF was detected, demonstrating that PrrF is implicated in an adaptive mechanism by which tobramycin promotes biofilm formation (Tahrioui et al., 2019). The involvement of sRNA PA0805.1 in the regulation of antibiotic fitness in P. aeruginosa was confirmed by observing the sensitivity of a mutant strain lacking PA0805.1 versus the wild-type strain to tobramycin under swarming conditions (Coleman et al., 2020). The sRNA Sr0161 and sRNA ErsA, interacting with oprD mRNA, lead to increased bacterial resistance to meropenem (Zhang et al., 2017). Pseudomonas aeruginosa magnesium transporter inhibits ExsA mediated T3SS gene transcription via the RsmA/RsmY/RsmZ signaling pathway (Chakravarty et al., 2017). The sRNA Sr006 is associated with polymyxin resistance (Zhang et al., 2017). When using azithromycin to treat infection, azithromycin exerts a bacteriostatic effect by indirectly inhibiting the transcription of rsmY and rsmZ by decreasing the expression of positive regulators of rsmY and rsmZ genes (Perez-Martinez & Haas, 2011). Ajoene, a sulfur rich molecule in garlic, exerts its QS inhibitory effect by regulating sRNA expression of rsmY and rsmZ in P. aeruginosa (Jakobsen et al., 2017). In conclusion, sRNAs exist in a variety of drug targets related investigations, therefore, some sRNAs are the promising candidates to become new antibiotic targets.

Conclusions

The sRNA is an indispensable part of the regulatory network of P. aeruginosa. It controls the expression of bacterial genes by regulating protein and target mRNA. The sRNA is transcribed under the stimulation of different environmental signals which usually does not need translate, so its response speed is faster than most proteins and mRNAs. The role of sRNA in post-transcriptional regulation has been identified, indicating their importance to the normal physiology and pathogenicity of P. aeruginosa. Current studies have revealed that sRNAs can regulate carbon / nitrogen / iron metabolism, biofilm formation, quorum sensing, drug resistance formation, virulence factor expression, and oxidative stress response of P. aeruginosa at the post-transcriptional level. To play their corresponding functions, most sRNAs need to form RNA-protein complexes with RNA chaperone Hfq. The newly discovered RNA chaperone ProQ increases the complexity of RNA-protein complexes regulating the metabolic networks (Gerovac et al., 2021). With the wide application of high-throughput sequencing technology, more and more sRNAs have been detected, but the further and more specific functions remain to be clarified. Bacterial sRNA is not only crucial to itself but also has an important impact on the host. They can be transferred to host cells through different mechanisms, affecting cell immune regulation, metabolism, and apoptosis, resulting in different consequences, such as sRNA transmitted through OMV (Diallo & Provost, 2020). The inherent and rapidly acquired resistance of P. aeruginosa to antibiotics is a challenging problem in clinical treatment. Due to the emergence of multidrug-resistant bacteria, new methods such as antibiotic-independent phage therapy and the use of antisense oligonucleotide peptide nucleic acid (PNA) to regulate gene expression have gradually appeared in people’s vision (Chevallereau et al., 2016; Perera, Carufe & Glazer, 2021). To find how sRNA plays an important role in the regulatory network or the pathogen-host interaction, clarifying the function of sRNA will be conducive to developing advance disease treatment strategies and promoting the search for new antibiotics and their action targets.

Additional Information and Declarations

Competing Interests

Author Contributions

Data Availability

The authors declare there are no competing interests.

Pei Liu conceived and designed the experiments, performed the experiments, analyzed the data, prepared figures and/or tables, authored or reviewed drafts of the article, and approved the final draft.

Changwu Yue performed the experiments, analyzed the data, prepared figures and/or tables, authored or reviewed drafts of the article, and approved the final draft.

Lihua Liu performed the experiments, analyzed the data, authored or reviewed drafts of the article, and approved the final draft.

Can Gao performed the experiments, analyzed the data, authored or reviewed drafts of the article, and approved the final draft.

Yuhong Lyu performed the experiments, analyzed the data, authored or reviewed drafts of the article, and approved the final draft.

Shanshan Deng performed the experiments, analyzed the data, authored or reviewed drafts of the article, and approved the final draft.

Hongying Tian performed the experiments, analyzed the data, prepared figures and/or tables, authored or reviewed drafts of the article, and approved the final draft.

Xu Jia performed the experiments, analyzed the data, prepared figures and/or tables, authored or reviewed drafts of the article, and approved the final draft.

The following information was supplied regarding data availability:

There is no raw data or code in this literature review.

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
