# Peer review of "The function of small RNA in Pseudomonas aeruginosa"

_PeerJ, doi:10.7717/peerj.13738_

## Round 0.1 · original submission · Major Revisions

Based on the reviewer comments, I feel that the review article has merit but needs more information before it can be considered for publication.

Reviewer 1 ·

Basic reporting

In this review authors focused on the functions of small RNAs in Pseudomonas aeruginosa.
Overall the data are presented clearly and are analyzed in an appropriate manner.
However few spelling corrections are needed
Pseudomonas aeruginosa needs to be in italic throughout the manuscript.
The authors did not discuss sRNAs containing CRISPR sequences
I would be nice if authors can include some figures to show the regulatory mechanism of the sRNAs
in line 18, it should be a gram-negative bacteria
In line 163: "The transcription of RsmY and RsmZ is positively regulated by RsmA and negatively
164 regulated by RsmY and RsmZ" not clear what the authors mean

Experimental design

n/a

Validity of the findings

n/a

Reviewer 2 ·

Basic reporting

See attached

Experimental design

See attached

Validity of the findings

See attached

Annotated reviews are not available for download in order to protect the identity of reviewers who chose to remain anonymous.

·

Basic reporting

The article has been fairly written stating the importance of small RNAs in P. aeruginosa. However there are quite a few questions/suggestions that has to be addressed before the article is considered for publication. These comments are mentioned below:

1. The English language usage used in the article should be assessed by a fluent English speaker. There are few areas in the text that need considerable improvement. For Instance, I would recommend in restructuring the abstract section. Lines 19-20: The sentence is not clear and the authors should consider restructuring.

2. The background should be elaborated. The Line 45: Here the authors should also mention about the co existence of P. aeruginosa with other bacteria in a biofilm. There are references missing in different parts of the manuscript.
(a) Line 49: reference missing
(b) Line 96: reference missing
(c) Line 149: Pseudomonas in italics
(d) Line 212: E. coli in italics
(e) Line 223: Pseudomonas in italics
(f) Line 231 : Pseudomonas in italics
(g)Line 313: reference missing
(h) Line 329 : reference missing
(i) Line 331: reference missing
(j) Line 356: Pseudomonas in italics, Yushan et al (date????)
(k) Line 362 : Pseudomonas in italics
(L) Line 365 : Pseudomonas in italics: Line 366: Transporter in small letter

Experimental design

1. In Survey methodology, Line 36: the authors have mentioned about some documents. They will have to elaborate on what type of documents did they search for.

2. Line 97: Here, the authors should describe AmrZ in 1-2 sentences with reference.
The section biofilm formation (Line 93) should be elaborated. Only one small RNA involved in biofilm formation has been stated here. The section should be elaborated to include almost all small RNAs involved in biofilm formation.

3. The same is applicable for sections including Quorum sensing (line 100) and Drug resistance (line 110). Both the sections should be elaborated with more references included. Additionally, I would suggest adding a picture/diagram or graphical representation of the sRNAs involved in quorum sensing.

4. Line 125, explain what is T3SS in few sentences. What is ExsA. Is it sRNA? If so explain. The construction of sentence is confusing.

5. Line 165: It is mentioned that the secondary structure of RsmY and RsmZ is similar. In this scenario, it would be better to include the picture or secondary structure of RsmY.

Validity of the findings

Nothing to add

---

## Round 0.2 · accepted · Accept

The authors have responded to all the reviewer comments. I recommend this for publication.

Reviewer 1 ·

Basic reporting

no comment

Experimental design

no comment

Validity of the findings

no comment

Additional comments

The authors have addressed all the comments. Overall the review now looks good. One minor comment is a figure showing how sRNAs in general function is missing.

Reviewer 2 ·

Basic reporting

Meets the standard.

Experimental design

Meets the standard.

Validity of the findings

Meets the standard.